# Liposomal Drug Delivery of *Blumea lacera* Leaf Extract: In-Vivo Hepatoprotective Effects

**DOI:** 10.3390/nano12132262

**Published:** 2022-06-30

**Authors:** Mohammad Hossain Shariare, Nusrat Jahan Khan Pinky, Joynal Abedin, Mohsin Kazi, Mohammed S. Aldughaim, Mohammad N. Uddin

**Affiliations:** 1Department of Pharmaceutical Sciences, North South University, Dhaka 1229, Bangladesh; mohammad.shariare@northsouth.edu (M.H.S.); nusrat.khan@nsu.edu.bd (N.J.K.P.); joynal.abedin@nsu.edu.bd (J.A.); 2Department of Pharmaceutics, College of Pharmacy, King Saud University, P.O. Box 2457, Riyadh 11451, Saudi Arabia; 3Research Center, King Fahad Medical City, Riyadh 11525, Saudi Arabia; maldughaim@kfmc.med.sa; 4College of Pharmacy, Mercer University, 3001 Mercer University Drive, Atlanta, GA 30341, USA

**Keywords:** liposome, *Blumea lacera*, hepatotoxicity, particle size, inflammations

## Abstract

Background: *Blumea lacera* (*B. lacera*) is a herbaceous plant commonly found in south-east Asia. It shows significant therapeutic activities against various diseases. The objectives of this study were to evaluate hepatoprotective effects of *Blumea lacera* leaf extract and also to investigate the comparative effectiveness between a liposomal preparation and a suspension of *B. lacera* leaf extract against carbon tetrachloride (CCl_4_)-induced liver damage. Methods: *B. lacera* leaf extract was characterized using a GC-MS method. A liposomal preparation of *B. lacera* leaf extract was developed using an ethanol injection method and characterized using dynamic light scattering (DLS) and electronic microscopic systems. The hepatoprotective effects of *B. lacera* leaf extracts and its liposomal preparation were investigated using CCl_4_-induced liver damage in Long Evan rats. Results: GC-MS data showed the presence of different components (e.g., phytol) in the *B. lacera* leaf extract. DLS and microscopic data showed that a liposomal preparation of *B. lacera* leaf extracts was in the nano size range. In vivo study results showed that liposomal preparation and a suspension of *B. lacera* leaf extract normalized liver biochemical parameters, enzymes and oxidative stress markers which were elevated due to CCl_4_ administration. However, a liposomal formulation of *B. lacera* leaf extract showed significantly better hepatoprotective effects compared to a suspension of leaf extract. In addition, histopathological evaluation showed that *B. lacera* leaf extract and its liposomal preparation treatments decreased the extent of CCl_4_-induced liver inflammations. Conclusion: Results demonstrated that *B. lacera* leaf extract was effective against CCl_4_-induced liver injury possibly due to the presence of components such as phytol. A liposomal preparation exhibited significantly better activity compared to a *B. lacera* suspension, probably due to improved bioavailability and stability of the leaf extract.

## 1. Introduction

The liver is a major internal organ which has a significant role in conjugation and purification of different chemical compounds and narcotics. The liver is susceptible to many diseases due to its strategic location, multi-dimensional functions, and vulnerability to a wide range of metabolic, toxic, microbial, neoplastic, and circulatory insults [1]. Hepatic fibrosis is caused by inflammation, abnormal buildup of extracellular matrix (ECM), and tissue remodeling during wound healing [2]. Liver cirrhosis is an inevitable phase in the process of liver injury. Hepatocellular carcinoma is positively associated with the occurrence of liver cirrhosis and chronic hepatitis [3,4]. Therefore, avoidance of hepatic inflammation and fibrosis is inevitable in preventing the manifestation and progression of liver cirrhosis and hepatocellular carcinoma. Several experiments have demonstrated that free radicals and reactive oxygen species (ROS) play a major part in inducing and independently controlling the progression of liver fibrosis [5,6]. Various xenobiotics such as carbon tetrachloride (CCl_4_) are known to cause hepatotoxicity. Since single administration of CCl_4_ promotes the development of many forms of ROS, steatosis and centric necrosis, chronic and repetitive administration contributes to liver fibrosis, cirrhosis, and ultimately hepatocellular carcinoma [7]. Hepatotoxicity induced by CCl_4_ includes development of free radicals that can accelerate the oxidative degradation of lipids. Some of the metabolic pathway end-products and enzymes which are very sensitive to abnormalities may be regarded as biochemical markers of hepatic dysfunction [8]. Liver disorders can be identified through liver function diagnosis and serum tests that can be measured by multiple biomarkers. Currently researchers are trying to find naturally occurring antioxidants (plants), which will prove beneficial in the prevention of oxidative damage [7].

Globally, around 25% of the prescribed drugs are produced from plant origins. There are wide ranges of medicinal plants found in nature whose various parts possess different pharmacological activities [9]. Around 80% of people in the world depend on traditional medicines obtained from plants for their primary health care requirements [10]. Recent widespread interest in plant-derived drugs indicates the importance of natural plant-based products in healthcare. It is also mandatory to prioritize the study of medicinal plants used in folklore in order to explore new plant-based medicine [11].

*B. lacera* is a herbaceous plant belongs to the Asteraceae family, commonly found in south-east Asia. *B. lacera* is used traditionally for its anti-diarrheal, antimicrobial, anxiolytic, anti-atherothrombosis, membrane stabilizing, anti-diabetic and alpha-amylase inhibitory activities. *B. lacera* has also been found to have antibiotic, antimicrobial and antioxidant effects [11,12,13]. A previous study showed that *B. lacera* leaf extract ameliorated acute ulceration and oxidative stress in an ethanol-induced rat model [14].

*Blumea balsamifera*, another herb of the family Asteracea, is a source of different volatile oils and flavanoids (e.g., l-borneol), and shows free radical scavenging, superoxide radical scavenging, antibacterial, anticancer, antifungal and anti-obesity effects [15]. *Blumea* flavanoids were found to have protective effects on liver injury [16,17,18,19,20]; however, to the best of our knowledge, no studies have been conducted on the performance of *B. lacera* leaf extract for CCl_4_-induced hepatotoxicity and oxidative stress in a rat model. Previous research studies suggest that phytoconstituents are poorly soluble and stable in biological systems, particularly when delivered through the oral route. The oral route of administration results in insufficient absorption and bioavailability for hydrophobic plant phytoconstituents [21]. There are numerous formulation techniques to solve the poor solubility, absorption, and low bioavailability problems of medicinal phytoconstituents, and nanotechnology is one of them. In order to provide controlled and targeted delivery, mitigate drug toxicity, increase bioavailability, and enhance patient compliance, liposomal encapsulated drugs have become a promising platform [21,22,23].

This study, therefore, was conducted to investigate the hepatoprotective activity of *B. lacera* leaf extracts against CCl_4_-induced hepatic injuries. A liposomal preparation of *B. lacera* leaf extract was developed and the therapeutic efficacy was compared between the liposomal preparation and a suspension of *B. lacera* leaf extract against CCl_4_-induced liver damage. The activity of the *B. lacera* liposomal preparation on inflammatory cell infiltration, and prevention of early-stage liver fibrosis in rat liver, were also investigated.

## 2. Materials and Methods

### 2.1. Materials

*B. lacera* plants were obtained from Gazipur District, Dhaka, Bangladesh. Egg phospholipid was used in this study prepared from egg yolk in-house. HPLC grade acetone, ethanol, dichloromethane and methanol were obtained from Sigma Aldrich (Darmstard, Germany). Human Biochemical and Diagnostic GmbH (Wiesbaden, Germany) kits were used for aspartate aminotransferase (AST), alanine aminotransferase (ALT), and alkaline phosphatase (ALP) assays. Carbon tetrachloride (CCl_4_), glacial acetic acid, thiobarbituric acid (TBA), concentrated hydrochloric acid, dimethyl sulfoxide (DMSO), sulfanilamide, naphthyl ethylene diamine dihydrochloride (NED), sodium hydroxide, potassium iodide, O-dianisidine, hydrogen peroxide (H_2_O_2_), and 5,5′-dithiobis-2-nitrobenzoic acid (DTNB), were obtained from Sigma Aldrich (Anklehwar, Gujrat, India).

### 2.2. Methods

#### 2.2.1. *B. lacera* Leaf Drying, Pulverization and Extraction Process

Fresh leaves of *B. lacera* were thoroughly washed using distilled water then dried in the shade for 12–15 days at room temperature. After drying, plant leaves were cut and ground to obtain a fine powder. The extraction process was performed using a maceration method in which 150 g of leaf powder was added with ethanol (solvent) in a conical flask. Then, the flask was kept on a rotary digital shaker for 6 days. After vigorous mixing, the liquid extract content was separated using filter cloth and then solvent was removed from the mixture using a rotary evaporator at 40 °C. The extract was kept at room temperature to solidify and was used in the preparation of suspension and liposomal preparation.

#### 2.2.2. Gas Chromatography Mass Spectroscopy (GC-MS) Method

The *B. lacera* leaf extract was analyzed using a Perkin Elmer (model Clarus 600 T, Waltham, MA, USA) mass spectrometer. An Elite 5MS column (30 m × 0.25 mm × 0.25 μm film thickness), at a flow rate of 1 mL/min helium carrier gas was used. A splitless injector at 20:1 was used, and the temperature was maintained at 280 °C. The temperatures of the MS ion source and inlet line were 200 °C and 220 °C, respectively. The GC-MS scan range was fixed at 40–600 mass ranges at a 70 eV energy level. Analysis time was 58 min for each sample.

#### 2.2.3. Preparation of Liposome of *B. lacera* Leaf Extract

The liposomal formulation of *B. lacera* leaf extract was prepared using an ethanol injection method. Egg phospholipid was first dissolved in ethanol and the extract was dissolved in phosphate buffer (pH 5.8). Then, the ethanolic solution of phospholipid was injected into phosphate buffer containing extract at rate of 1 mL/s. These two solutions were mixed for 10 min at 1000 rpm. After 10 min, ethanol was evaporated from the mixer at around 35 °C. The liposomal preparation of *B. lacera* leaf extract was then characterized using dynamic light scattering and microscopic methods.

#### 2.2.4. Dynamic Light Scattering (DLS) Method

The particle size distribution and zeta potential value of the liposomal preparation of the *B. lacera* leaf extract were determined using a Malvern zetasizer ZS90 (Malvern Instrument, Malvern, UK) at 25 °C, where each batch of the liposomal preparation was characterized in triplicate.

#### 2.2.5. Transmission Electron Microscopy (TEM)

Transmission electron microscopy TEM (Jeol JEM1010, Akishima, Japan) was used for the morphological investigation of liposomal batches of *B. lacera* leaf extract. Dried and coated samples were loaded on the TEM and viewed at 5000–100,000 magnifications.

#### 2.2.6. Entrapment Efficiency Method

Entrapment efficiency of the liposomal preparation was determined for the optimized batch of *B. lacera* leaf extract. The liposomal preparation of *B. lacera* was centrifuged at 10,000 rpm (Smart R17 micro refrigerated centrifuge, Hanil science industrial, Korea) equivalent to 14,437 g force for 30 min at 4 °C. Supernatant was separated from the solid content and the solid content was washed twice using phosphate buffer (pH 5.8). Entrapment efficiency was determined after lysis of liposomes with ethanol followed by sonication for 15 min. The solution was then diluted with buffer (pH 5.8) and the absorbance of encapsulated extract in the diluted buffer solution was determined in triplicate at 241 nm using a UV–vis spectrophotometer [24]. The concentration of the encapsulated and free drug in the supernatant was determined by a calibration curve using phytol as a standard in a concentration range of 10 µg–100 µg. The following formula was used to determine the encapsulation efficiency:Encapsulation efficiency (%) = (Entrapped drug/Total drug) × 100
where, Total drug = Free drug in supernatant + encapsulated drug.

#### 2.2.7. Preparation of *B. lacera* Suspension

The standard *B. lacera* suspension was prepared by mixing *B. lacera* leaf extract with normal saline solution (0.9% NaCl). Each milliliter of suspension contained 8 mg of extract.

#### 2.2.8. In-Vivo Hepatotoxicity and Oxidative Stress Study

##### Animals

Thirty-five Long-Evans rats (180–250 g), age 9–12 weeks, were obtained from the animal house at the Department of Pharmaceutical Sciences, North South University. Groupwise they were separated in different cages at room temperature (25 °C) with 12-h dark/light cycles. The experimental procedure was reviewed and approved by the institutional animal ethics research committee (project identification code- AEC-016–2018, approved on 27 November 2018) at North South University, Dhaka, Bangladesh.

##### Design of Experiment

Thirty-five Long-Evan rats were divided into five groups to evaluate the hepatoprotective effect of *B. lacera* leaf extract; each group contained seven rats. The groups were control (oil), control (blank liposome), disease group, treatment group 1 using *B. lacera* leaf extract suspension and treatment group 2 using liposomal preparation of *B. lacera* leaf extract.

The control group (olive oil) received (1 mL/kg body weight) olive oil administered intragastrically every day for fourteen days. The control group (blank liposome) received (1 mL/kg body weight) blank liposome suspension administered intragastrically to the rodents for fourteen days. The blank liposome suspension was comprised of egg phospholipid and poloxamer 407 (stabilizer, prepared by the ethanol injection method. The disease group received CCl_4_ as a disease inducer at a 1:3 ratio (CCl_4_: olive oil) (1 mL/kg body weight) intragastrically twice weekly for fourteen days. Treatment group 1 (suspension of *B. lacera* leaf extract and CCl_4_) were treated with CCl_4_ at a ratio of 1:3 (CCl_4_: olive oil) (1 mL/kg body weight) intragastrically two times weekly for fourteen days. The group was also given *B. lacera* leaf extract suspension at a dosage of 25 mg/kg body weight (8 mg/mL suspended in saline) for fourteen days every day. The treatment group rats were fed CCl_4_ (1:3 CCl_4_ in olive oil) for fourteen days at a dose of 1 mL/kg body weight, twice a week intragastrically. This group was also treated intragastrically with a liposomal nano formulation of *B. lacera* leaf extract at a dosage of 25 mg/kg body weight (8 mg extract in per ml of liposomal preparation) every day for fourteen days.

##### Preparation of Tissue and Plasma Samples

Animals were observed for fourteen days, then all rats were sacrificed. Ketamine and pentobarbitone were used to anesthetize the rats. Blood and liver samples were collected, and some other organs were also immediately collected. These samples were then weighed and stored at −20 °C for further studies. Blood serum (plasma) and liver homogenates were centrifuged at 8000 rpm for 15 min at 4 °C. The transparent plasma and liver supernatants were then collected using a micropipette with Eppendorf tubes and subsequently preserved at −20 °C. These samples were later used in further analyses of protein and enzyme determination.

##### Analysis of Hepatotoxicity

Estimation of liver Biomarker Enzymes

Liver biomarkers aspartate aminotransferase (AST), alanine aminotransferase (ALT), and alkaline phosphatase (ALP) were estimated in plasma by following the manufacturer’s protocols (Human Biochemical und Diagnostic GmbH assay kits, Wiesbaden, Germany) [25].

Evaluation of Oxidative Stress Markers

Lipid peroxidation occurs due to production of free radicals in organisms. Over exposure of free radicals increases malondialdehyde (MDA) production, thus MDA level is generally known as an oxidative stress marker [26]. Lipid peroxidation was calorimetrically calculated in the liver by estimating TBA reactive substances. Absorbance was taken at 535 nm against a reference blank.

Nitric oxide (NO) was calculated as nitrate according to the method described by Tracey et al. [27]. In this analysis naphthyl ethylene diamine dihydrochloride (NED) was used at 0.1% *w*/*v*. Absorbance was measured against the blank solutions at 540 nm. NO level is expressed as tissue nmol/mL and NO amount was determined using a standard curve.

Modified Witko-Sarsat et al. [28] and Tiwari et al. [29] methods were used to determine advanced protein oxidation product (APOP) levels in the plasma and liver homogenates. A volume of 10 µL of sample was diluted with phosphate buffer saline at a 1:5 ratio. After 2 min, 50 µL acetic acid and 50 µL of potassium iodide were added into each sample. Absorbance of the reaction mixture was immediately recorded at 340 nm.

Estimations of Liver Antioxidant Enzymes

Catalase (CAT) activities were estimated using the method described by Khan [30]. The reaction solution of CAT activities consisted of phosphate buffer (2.5 mL, 50 mmol), enzyme extract (0.1 mL), and hydrogen peroxide (0.4 mL). Absorbance changes of the reaction solution were determined at 240 nm. Superoxide dismutase (SOD) activity was determined in plasma and tissue homogenates using the method described by Mishra et al. [31] Absorbance changes were recorded for 1 min at 480 nm at 15 s intervals Glutathione (GSH, g-glutamylcysteinylglycine) level was analysed according to the method of previous studies [32,33,34], where absorbance changes were measured at 412 nm. The myeloperoxidase (MPO) bioassay was done in 96-well plates using a modified dianisidine-H_2_O_2_ method [32]. Tissue homogenate samples (10 microgram protein) were mixed in triplicate. Then, the sample mixture was added to o-dianisidine dihydrochloride (Sigma), potassium phosphate buffer (pH 6.0) and hydrogen peroxide and absorbance changes were estimated at 412 nm.

Histopathological Study

Liver tissues were immediately collected after sacrifice of rats for microscopic evaluation. The tissues were kept in a 10% buffered formaldehyde solution for 24 h before histopathological analysis. Samples were then fixed into paraffin wax, and 5 μm sections were made. Hematoxylin and eosin (H&E) stain was used to stain sections and then these were placed on microscope glass slides to test liver inflammation, hepatocyte ballooning, and necrosis. This study was performed at a magnification of around 40× with a florescence light microscope.

#### 2.2.9. Statistical Analysis

Data obtained in this study were expressed as mean ± standard error of mean (SEM). GraphPad Prism 9.2.0 version (San Diego, CA, USA) was used to conduct one-way analysis of variance (ANOVA) tests; the level of significance *p* ≤ 0.05 was considered statistically significant. Normality tests are all biomarkers and Non-parametric tests (Kruskal-Wallis test) for some biomarkers are also performed (Appendix A).

## 3. Results

### 3.1. GC-MS Study of B. lacera Leaf Extract

GC-MS data suggest that major components present in the leaf extract of *B. lacera* were hexadecanoic acid, 2-hexadecan-1-OL (Phytol), neophytadiene, and 1, 2-epoxyundecane, among others (Figure 1 and Table 1). Previous research showed that phytol exhibits different pharmacological activity including cytotoxic, hepatoprotective, antiinflammatory and antioxidant activity [24]. A literature review of *B. lacera* leaf extract also showed that it has strong antiinflammatory and antioxidant effects, which may be due to the presence of phytol. Phytol is poorly soluble in water, exhibits poor absorption from the intestine and is unstable at acidic pH. Previous studies have suggested that a nano carrier-based drug delivery system may be able to enhance the solubility and stability of the encapsulated drugs [24]. Therefore, in this study, a liposomal drug delivery system was used to enhance solubility and bioavailability and provide better physico-chemical stability.

### 3.2. Particle Size Distribution, Zeta Potential and Entrapment Efficiency

Particle size distribution data determined using the dynamic light-scattering method showed that the optimized liposomal batches of *B. lacera* leaf extract had average particle sizes in the range of 122–135 nm and polydispersity index (PDI) values of 0.10–0.19 (Table 2). Transmission electron microscopic (TEM) data showed similar results; polygonal vesicles (liposomes) were observed, mostly less than 150 nm in size (Figure 2). Liposomes with polygonal shape-like morphology were observed by Johnsson and Edwards [35] using cryo TEM. The Zeta potential values measured for these liposomal batches were −9–11.6 mV, with entrapment efficiency of 80–83% (Table 2).

### 3.3. Effect of B. lacera Leaf Extract and Its Liposomal Preparation on Body Weight, Food, and Water Intake

The bodyweights of all Long Evan rats were measured every fourteen days during the experiment, and weight changes were determined for all groups. It was noticed that in those fourteen days, all the rats gained some weight. However, comparing the two control groups, the disease group exhibited more weight gain. Group 4, which was treated with the suspension of *B. lacera* leaf extract, showed intermittent weight gain, while the rat group treated with the liposomal preparation of *B. lacera* leaf extract showed a major weight gain. The CCl_4_-intoxicated group showed reduced intake of food and water compared to the control groups (Figure 3, Figure 4 and Figure 5 and Table 3). Food and water consumption was improved in rats of group 4 and group 5 (treatment groups) relative to the CCl_4_-intoxicated group after 5 days.

### 3.4. Effect of B. lacera Leaf Extract and Its Liposomal Preparation on Different Organ Wet Weights

*B. lacera* showed marked effects on the CCl_4_-treated rat’s organ weights observed from the wet weight of the organs (Table 4). CCl_4_-treated rats showed a reduction in wet weight of the liver compared to control group rats. Results showed that Long Evan rats treated with the liposomal preparation of *B. lacera* leaf extract improve liver wet weight more compared to rat groups treated with *B. lacera* suspension. However, the weights of brain, heart, and lung were not improved markedly for CCl_4_-intoxicated rat groups treated with *B. lacera* leaf extract suspension and its liposomal preparation. The wet weight of kidneys did not show any major differences between control groups, the disease group and rat groups treated with *B. lacera* suspension and liposomal preparation. The spleen wet weight of the disease group was increased significantly compared to control groups, and was later decreased markedly when treated with *B. lacera* leaf extract suspension and its liposomal preparation.

### 3.5. Effect of B. lacera Leaf Extract and Its Liposomal Preparation on Liver Function Biomarkers

We determined the level of liver function biomarkers (ALT, AST, ALP) in plasma serum samples to confirm the toxic effect of CCl_4_. Biochemical liver function experiments showed that the introduction of CCl_4_ has significantly increased the plasma biomarkers level relative to control rats (*p* ≤ 0.05) (Figure 6). Rat groups treated with *B. lacera* leaf extract suspension and its liposomal preparation significantly reduced the increased biomarkers levels in plasma serum compared to the disease group (CCl_4_ group) (Figure 6). However, the rat group treated with liposomal nano formulation of *B. lacera* leaf extract showed significant reduction in liver biomarker levels compared to the rat group treated with the *B. lacera* suspension (*p* ≤ 0.05) (Figure 6).

### 3.6. Effect of B. lacera Leaf Extract and Its Liposomal Formulation on Oxidative Stress Markers

Several forms of reactive oxygen species (ROS) are generated after metabolism of CCl_4_ in the liver, and may increase lipid oxidative degradation. Therefore, in this study increased oxidative stress and lipid peroxidation parameters were estimated, and the quantities of malondialdehyde (MDA), advanced protein oxidation product (APOP), and nitric oxide (NO) in liver homogenates and plasma serum were investigated to assess oxidative stress.

In the CCl_4_-intoxicated group, the lipid peroxidation product MDA concentration in both liver homogenates and plasma serum was significantly elevated relative to the control groups (Figure 7). However, rat groups treated with *B. lacera* leaf extract and its liposomal preparation reduced the elevated level of MDA significantly (Figure 7). In addition, the rat group treated with the liposomal preparation of *B. lacera* leaf extract reduced MDA level more significantly compared to the *B. lacera* leaf extract suspension treatment group (Figure 7). APOP concentration level was increased extensively in plasma and liver in CCl_4_-intoxicated rats compared to control groups rats. Rat groups treated with *B. lacera* leaf extract and its liposomal nano formulation (Figure 7) reduced the APOP level significantly both in liver and plasma, while the *B. lacera* liposomal nano formulation showed better activity in APOP level reduction, and particularly in plasma serum level (Figure 7). NO level was substantially elevated in both plasma and liver homogenates in CCl_4_-intoxicated group rats relative to control groups (Figure 7). The rat group treated with the *B. lacera* nano formulation reduced the elevated NO amount in the plasma and liver homogenates significantly compared to *B. lacera* suspension group rats (*p* ≤ 0.05) (Figure 7).

### 3.7. Effect of B. lacera Leaf Extract and Is Liposomal Preparation on Liver Antioxidant Enzymes

Antioxidant defense levels in tissue can be decreased with an increase of oxidative stress. Therefore, antioxidants level of the tissue such as SOD, GSH, and CAT activities were evaluated in this study. In the CCl_4_-intoxicated disease group, substantial reductions in antioxidants were found compared to the control groups (*p* ≤ 0.05) (Figure 8). Application of *B. lacera* leaf extract suspension and its nano preparation restored the activities of SOD, GSH, and CAT in CCl_4_-intoxicated rats. The *B. lacera* nano preparation also showed significant improvement in anti-oxidant levels compared to the *B. lacera* leaf extract suspension treatment group (Figure 8).

### 3.8. Effect of B. lacera Leaf Extract and Its Liposomal Preparation on Myeloperoxidase (MPO) Activity in Liver Tissue

Myeloperoxidase (MPO) activities in liver tissues were measured to assess liver inflammation and liquid permeation in liver cells. MPO activity increased significantly in the liver of CCl_4_-intoxicated rats compared to control groups rat, which was significantly decreased when treated with *B. lacera* leaf extract (*p* ≤ 0.05) and its liposomal preparation (*p* ≤ 0.001) (Figure 9). In this case, similarly, the *B. lacera* nano liposomal preparation treatment group showed significantly better normalization of MPO (*p* ≤ 0.05) compared to the *B. lacera* leaf extract suspension treatment group (Figure 9).

### 3.9. Effect of B. lacera Leaf Extract and its Liposomal Preparation on CCl_4_-Induced Histopathological Changes in the Liver

Standard morphological patterns of liver tissue were found in the control groups (Figure 10A,B). However, in the CCl_4_-intoxicated group samples stained with hematoxylin and eosin, histopathological changes such as inflammatory cell infiltration, hepatocyte ballooning into the portal tract, and extreme cellular necrosis in the liver tissue were observed and marked using the white arrow in Figure 10C. Treatments with *B. lacera* leaf extract (25 mg/kg) and its liposomal preparation recovered the CCl_4_-induced hepatic damages (Figure 10D,E). The antioxidative and hepatoprotective properties of *B. lacera* leaf extract probably reduced the inflammatory infiltrates, ballooning, and cellular necrosis in the liver tissue sections of rats. Histological study also showed that the liposomal preparation of *B. lacera* leaf extract recovered the liver injuries more than the *B. lacera* leaf extract suspension (Figure 10D,E). This result is also supported by the MPO activity studies performed for the treatment groups (Figure 9).

## 4. Discussion

This research was conducted to ascertain the therapeutic activity of *B. lacera* leaf extract on liver injury induced by CCl_4_, since previous studies suggest that *B. lacera* is an antioxidant-rich plant [12,13]. We also compared the in-vivo protective effect between *B. lacera* leaf extract suspension and a liposomal formulation prepared from *B. lacera* leaf extract. CCl_4_ is a xenobiotic, can alter hepatic characteristics and is widely used in animal models to induce hepatotoxicity. Oxidative stress is the main reason for hepatic damage induced by CCl_4_ [36]. Preventive effects on liver inflammation, fibrosis, cirrhosis, and hepatocellular carcinoma have been confirmed for different natural antioxidants [37].

CCl_4_-mediated liver damage was confirmed in the current study by the identification of a rapid elevation of hepatobiliary serum biomarker enzymes (ALT, AST, and ALP). The increased serum enzyme levels indicate mitochondrial damage, cellular disintegration, hepatocyte necrosis and functional damage of the cell membrane of the liver [38]. *B. lacera* leaf extract treatment showed substantial reduction in elevated biomarker enzymes level observed in CCl_4_-intoxicated rats. Results showed that rat groups treated with a liposomal preparation of *B. lacera* leaf extract improved liver damage more significantly compared to those treated with a *B. lacera* leaf extract suspension. The protective effect of *B. lacera* leaf extract probably related to the free radical scavenging activity of this plant [19,38], possibly related to the presence of components such as phytol, found during GC-MS analysis in this study. Previous studies suggest that phytol has strong anti-oxidant, anti-inflammatory and hepatoprotective activities [24]. The liposomal preparation probably resulted in improved solubility, permeability and bioavailability of this plant extract, which improved the anti-oxidant activity more significantly compared to *B. lacera* leaf extract suspension.

Previous investigations suggests that increased hepatocyte oxidative stress biomarkers occur probably due to the development of lipid peroxides by CCl_4_ free radical derivatives, which is one of the key reasons for hepatotoxicity. Antioxidant activity or inhibition of the formation of the free radicals is among the factors that may prevent hepatotoxicity and decrease the level of oxidative stress biomarkers induced by CCl_4_. A previous study also showed that a high level of NO is associated with acute liver damage caused by CCl_4_, which may raise the risk of liver cirrhosis [39]. In this study, a significant increase in oxidative stress marker levels was observed in the CCl_4_-intoxicated rat group relative to the control group. The CCl_4_-intoxicated rat group treated with *B. lacera* leaf extract and its liposomal preparation showed significant normalization of oxidative stress marker levels, probably through free radical scavenging activity that was elevated after CCl_4_ administration. These findings support the results of a previous study, which suggested that antioxidant-rich food or supplements can stop protein oxidation and lower oxidative stress biomarker concentrations in tissues and plasma of CCl_4_-administered rats [20].

Organisms acquire antioxidant mechanisms to inhibit oxidative stress, including antioxidant enzymes (SOD and CAT), and a non-enzymatic mechanism (GSH), which functions throughout the system to control unregulated oxidation cascades and protect cells from oxidative harm by scavenging reactive oxygen species [39,40]. Decreased antioxidant protection of the tissue results directly from elevated lipid peroxidation and oxidative stress. Antioxidants have been shown to substantially alleviate persistent oxidative stress conditions in the liver of CCl_4_-intoxicated rats [41]. SOD is the major antioxidant enzyme that protects cells and tissues from endogenous and exogenous generated ROS. CAT is a heme-containing enzyme and protects liver cells from oxidative damage by converting H_2_O_2_ into water and O_2_. On the other hand, the core extracellular redox regulator is GSH, which may directly purify ROS or free radicals by scavenging the free radicals or becoming part of the glutathione redox mechanism that requires glutathione reductase and glutathione peroxidase [21,39]. Compared with the control group, in this study substantial declines in liver and plasma SOD, CAT and GSH levels were observed in rats intoxicated with CCl_4_. The level of SOD, CAT and GSH were elevated significantly when treated with *B. lacera* leaf extract and its liposomal preparation. Results suggest that flavoproteins, such as glutathione reductase, had direct free radical scavenging activity that could have improved the antioxidant activity of *B. lacera* leaf extract and its liposomal preparation. Liposomal preparation of *B. lacera* leaf extract showed significantly better antioxidant effects than the *B. lacera* leaf extract suspension in every case.

Oxidative stress and lipid peroxidation-facilitated liver cell damage may produce an inflammatory response, which has been detected by infiltrating inflammatory cells alongside the blood vessels in the liver [7,20]. Earlier studies suggested that the invading inflammatory cells in CCl_4_-intoxicated rats are mostly monocytes and neutrophils that cause liver damage [42]. The metabolism of CCl_4_ controls the production of free radicals, which also stimulates the activation of Kupffer cells that release various inflammatory regulators such as interleukins and alpha-tumor necrosis factor (TNF-alpha) [43]. These cytokines cause several types of inflammation such as macrophage activation, synthesis of prostaglandins, and the infiltration of neutrophils in the inflamed area [44]. Our study also showed that inflammatory cell infiltration developed in CCl_4_-intoxicated rats. Our observations support the discovery that in liver homogenates of CCl_4_-treated rats MPO activity is amplified. This was reduced by *B. lacera* leaf extract and its liposomal preparation treatments. MPO is a major enzyme that is released after entry and activation of neutrophils, and catalyzes the development of hypochlorous acid and additional oxidizing species [45].

After biochemical assessments, *B. lacera* leaf extract and its liposomal preparation treatments were also evaluated by a histopathological study. Results showed that *B. lacera* leaf extract and its liposomal preparation treatments normalized the hepatic lesions produced by CCl_4_ administration. Sections of liver from treatments with *B. lacera* leaf extract and its liposomal preparation mostly exempt lacked inflammatory infiltrates, and cellular necrosis. Our in-vivo results suggest that the antioxidant, hepatoprotective, and anti-inflammatory properties of *B. lacera* leaf extract and its liposomal preparation normalized the systemic and biochemical levels of liver function significantly.

## 5. Conclusions

GC-MS analysis of *B. lacera* leaf extracts showed components, including phytol, which have been found to have antioxidant, hepatoprotective, and anti-inflammatory activities. Microscopic and dynamic light scattering data demonstrated that polygonal-like liposomal vesicles had an average size of less than 150 nm. In this study, *B. lacera* leaf extracts containing phytol and its liposomal preparation treatments significantly normalized elevated levels of liver biomarkers, antioxidant enzymes, and oxidative stress markers, and provided the liver with adequate protection against lipid peroxidation. However, the liposomal preparation exhibited significantly better hepatoprotection than the *B. lacera* leaf extract suspension. This was probably related to better solubility, permeability, and bioavailability of the leaf extract when delivered through a liposomal nano drug delivery system. These results suggest that a lipososmal nano drug delivery system might be a suitable platform to treat diseases such as liver cirrhosis, liver fibrosis, and chronic hepatic inflammation, using natural phytoconstituents.

## Figures and Tables

**Figure 1 nanomaterials-12-02262-f001:**
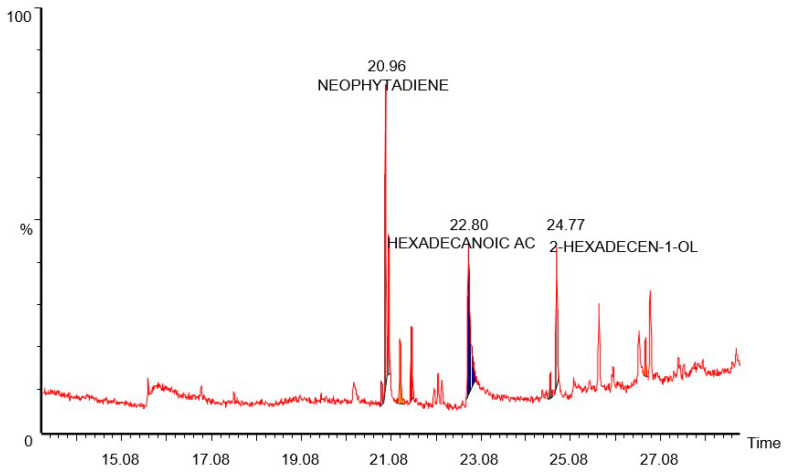
GC-MS chromatogram of *B. lacera* leaf extract.

**Figure 2 nanomaterials-12-02262-f002:**
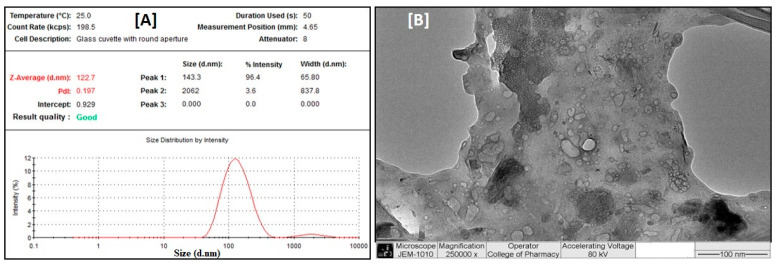
Particle size distribution (**A**) and TEM data (**B**) of liposomal preparation of the *B. lacera* leaf extract.

**Figure 3 nanomaterials-12-02262-f003:**
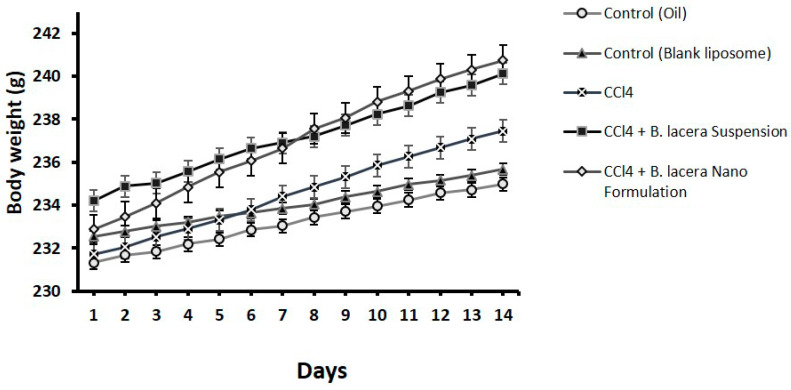
Effect of *B. lacera* leaf extract and its liposomal preparation on body weight of CCl_4_ treated rat for14 days. Data are presented as mean ± standard error of mean, where *n* = 6.

**Figure 4 nanomaterials-12-02262-f004:**
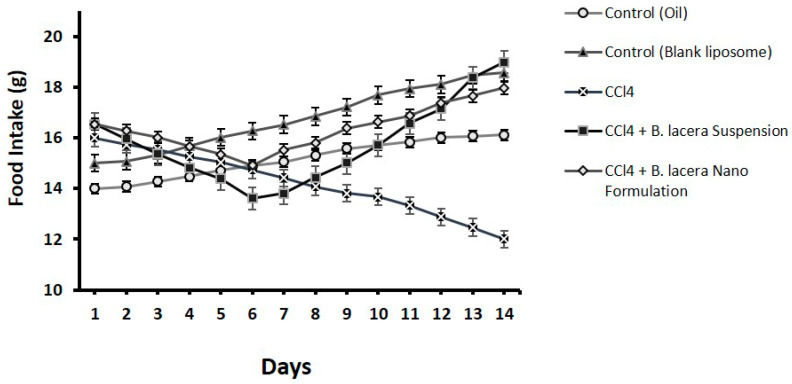
Effect of *B. lacera* leaf extract and its liposomal preparation on food intake of CCl_4_-treated rats for14 days. Data are presented as mean ± standard error of mean, where *n* = 6.

**Figure 5 nanomaterials-12-02262-f005:**
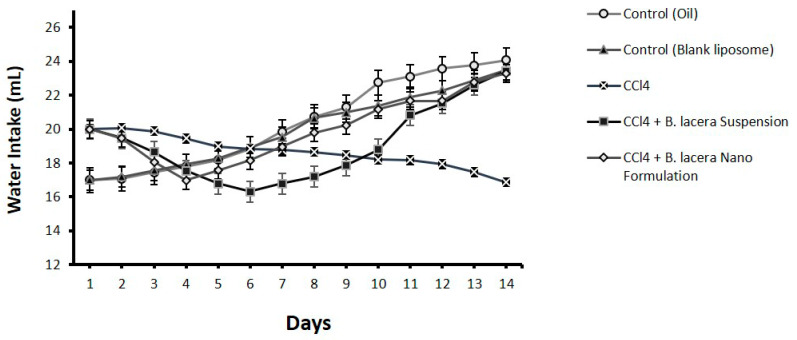
Effect of *B. lacera* leaf extract and its liposomal preparation on water intake of CCl_4_-treated rats for14 days. Data are presented as mean ± standard error of mean, where *n* = 6.

**Figure 6 nanomaterials-12-02262-f006:**
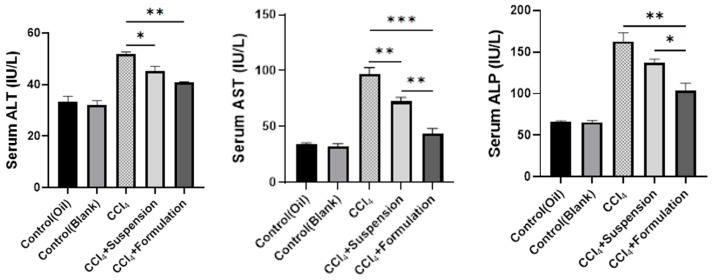
Effect of *B. lacera* leaf extract and its liposomal preparation on liver biomarker levels. Data are presented as mean ± SEM, *n* = 6. where * *p* ≤ 0.05, ** *p* ≤ 0.01, *** *p* ≤ 0.001, when compared between the CCl_4_, CCl_4_+ suspension and CCl_4_+ formulation (liposomal) groups. *p* ≤ 0.05 was considered statistical level of significance for all cases.

**Figure 7 nanomaterials-12-02262-f007:**
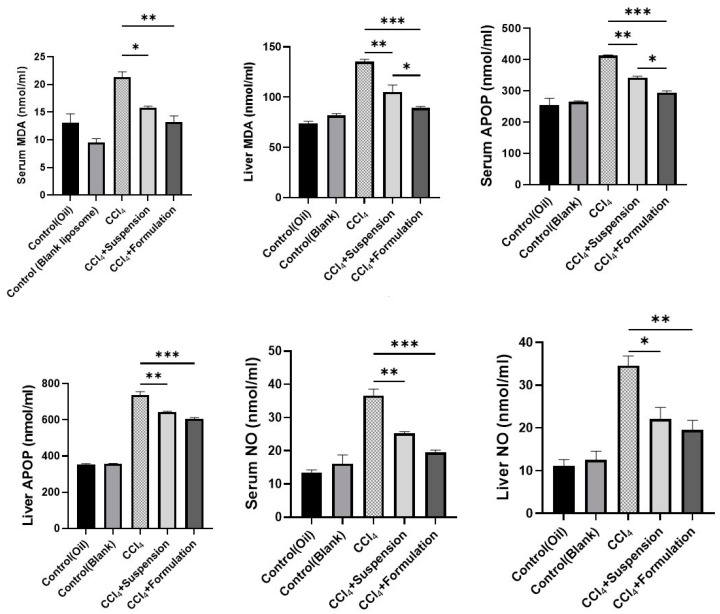
Effect of *B. lacera* leaf extract and its liposomal preparation on oxidative stress markers. Data are presented as mean ± SEM, *n* = 6. where * *p* ≤ 0.05, ** *p* ≤ 0.01, *** *p* ≤ 0.001, when compared between the CCl_4_, CCl_4_+ suspension and CCl_4_+ formulation (liposomal) groups. *p* ≤ 0.05 was considered statistical level of significance for all cases. MD: Malondialdehyde, APOP: Advanced protein oxidation products, NO: Nitric oxide.

**Figure 8 nanomaterials-12-02262-f008:**
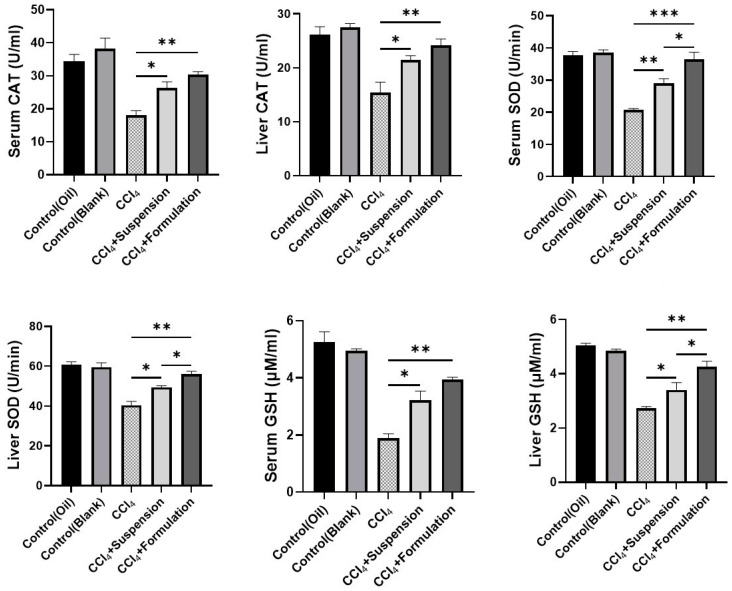
Effects of *B. lacera* leaf extract and its liposomal preparation on liver antioxidant enzymes. Data are presented as mean ± SEM, *n* = 6. where * *p* ≤ 0.05, ** *p* ≤ 0.01, *** *p* ≤ 0.001, when compared between the CCl_4_, CCl_4_+ suspension and CCl_4_+ formulation (liposomal) groups. *p* ≤ 0.05 was considered the statistical level of significance for all cases. CAT: Catalase, GSH: Glutathione, SOD: Superoxide dismutase.

**Figure 9 nanomaterials-12-02262-f009:**
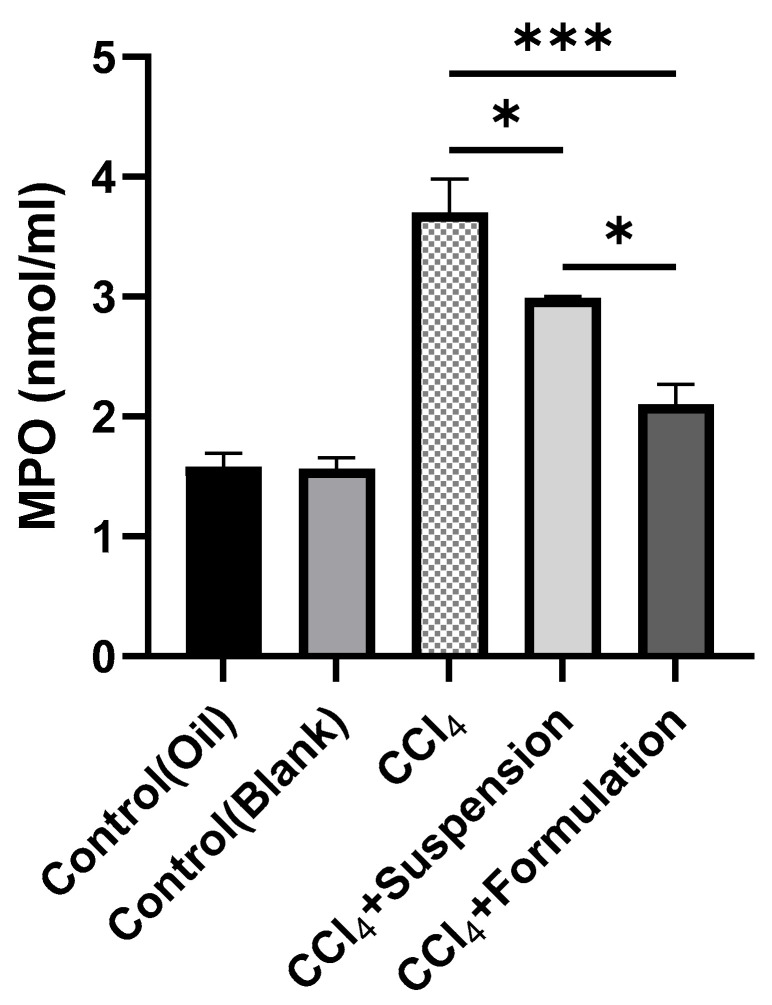
Effects of *B. lacera* leaf extract and its liposomal preparation on myeloperoxidase (MPO) activity. Data are presented as mean ± SEM, *n* = 6. where * *p* ≤ 0.05, *** *p* ≤ 0.001, when compared between the CCl_4_, CCl_4_+ suspension and CCl_4_+ formulation (liposomal) groups. *p* ≤ 0.05 was considered statistical level of significance for all cases.

**Figure 10 nanomaterials-12-02262-f010:**
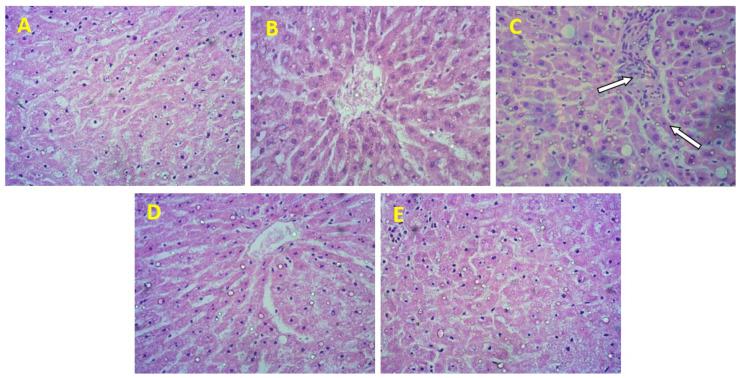
Effects of *B. lacera* leaf extract and its liposomal preparation on CCl_4_-induced hepatic inflammation. (**A**) Control (Oil), (**B**) Control (Blank), (**C**) CCl_4_, (**D**) CC_4_ + suspension of *B. lacera* leaf extract, (**E**) CCl_4_ + liposomal nano formulation of *B. lacera* leaf extract. *n* = 6 per group; magnification = 40×.

**Table 1 nanomaterials-12-02262-t001:** Major components of *B. lacera* leaf extract.

No.	Name	RT	Area%
1	Heptanal	20.88	2.040
2	Neophytadiene	20.96	17.770
3	1,2-Epoxyundecane	21.03	16.150
4	2-Dodecen-1-OL	21.28	6.700
5	Hexadecanoic Acid	22.80	22.950
6	Caryophyllene Diepoxide	24.62	2.320
7	2-Hexadecan-1-OL (Trans Phytol)	24.77	22.840
8	Ecosylacetate	26.75	2.380

**Table 2 nanomaterials-12-02262-t002:** Average size, PDI, Zeta potential and entrapment efficiency of three optimized batches of liposomes prepared using similar conditions. Data are presented as mean ± standard error of mean, where *n* = 6.

Batch No.	Average Particle Size (nm) ± SD	PDI ± SD	Zeta Potential (mV) ± SD	Entrapment Efficiency (%) ± SD
OPB 1	122.7 ± 2.5	0.19 ± 0.04	−10.7 ± 2.3	81.0 ± 2.5
OPB 2	133.3 ± 3.3	0.10 ± 0.03	−8.7 ± 3.2	80.0 ± 2.0
OPB 3	135.5 ± 4.1	0.12 ± 0.02	−11.6 ± 3.1	83.5 ± 3.0

**Table 3 nanomaterials-12-02262-t003:** Effect of *B. lacera* leaf extract and its liposomal preparation on body weight, food, and water intake of CCl_4_ treated rats for 14 days. Data are presented as mean ± standard error of mean, where *n* = 6.

Parameters	Control (Oil)	Control (Blank Liposome)	CCl4	CCl4 + *B. lacera* Suspension	CCl4 + *B. lacera* Nano Formulation
Initial Body Weights (g)	231.33 ± 0.23	232.54 ± 0.17	231.73 ± 0.36	234.21 ± 0.37	232.87 ± 0.57
Final Body Weight (g)	234.98 ± 0.21	235.68 ± 0.57	237.45 ± 0.35	240.12 ± 0.36	240.75 ± 0.44
Food Intake (g/day)	15.12 ± 0.21	16.77 ± 0.34	14.21 ± 0.33	15.76 ± 0.43	16.35 ± 0.24
Water Intake (mL/day)	20.38 ± 0.71	19.99 ± 0.59	18.69 ± 0.25	19.12 ± 0.60	19.98 ± 0.51

**Table 4 nanomaterials-12-02262-t004:** Effect of *B. lacera* leaf extract and its liposomal preparation on different organ wet weights. Data are presented as mean ± standard error of mean, where *n* = 6.

Different Wet Organs	Control (Oil)	Control (Blank)	CCl4	CCl4 + *B. lacera* Suspension	CCl4 + *B. lacera* Nano Preparation
Liver Wet Weight (g)	8.37 ± 0.40	8.51 ± 0.43	7.14 ± 0.31	7.42 ± 0.29	7.98 ± 0.34
Brain Wet Weight (g)	1.80 ± 0.19	1.81 ± 0.15	1.70 ± 0.16	1.71 ± 0.12	1.73 ± 0.10
Heart Wet Weight (g)	1.21 ± 0.08	1.15 ± 0.10	1.06 ± 0.05	1.07 ± 0.08	1.08 ± 0.06
Kidney Wet Weight (g)	1.12 ± 0.06	1.16 ± 0.08	1.06 ± 0.04	1.09 ± 0.05	1.11 ± 0.06
Lung Wet Weight (g)	1.66 ± 0.09	1.71 ± 0.11	1.55 ± 0.08	1.58 ± 0.06	1.61 ± 0.07
Spleen Wet Weight (g)	0.80 ± 0.05	0.69 ± 0.03	1.23 ± 0.07	0.58 ± 0.04	0.72 ± 0.05

## Data Availability

The data presented in this study are available on request from the corresponding author.

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
