# Peer review of "Liposomal Drug Delivery of Blumea lacera Leaf Extract: In-Vivo Hepatoprotective Effects"

_nanomaterials, 2022, doi:10.3390/nano12132262_

Round 1

Reviewer 1 Report

The present manuscript presents a study on the protective effect of B. lacera extract formulated as a liposomal form or a as suspension on liver induced injury in the rat. The results are well described and the findings are very interesting. Some details should, however, be corrected before publication:

Title: very confuse: Please correct. Suggestion: Liposomal Drug Delivery of Blumea lacera Leaf Extract: In-vivo hepatoprotective effects

Authors use the designation “liposomal nano preparations”. Liposomes used for therapeutic purposes are of nanometric scale, thus, remove “nano” after “liposomal” throughout the text.

Abstract: Blumea lacera: once abbreviated use such form throughout the text.

Ln 16: “…liposomal nano preparation of…”, please complete

Ln 21: use “CCl4 induced liver damage in rat” instead of “Long Evan rat model”

Use CCl4 instead CCl4

Ln 53: “biotransformation of free - radical that”, please confirm the idea

Ln 75, 450, 486: remove underlining

Ln 76, 87-91, 486: correct font size

Ln 106-107: please identify the chemicals supplier

Ln 110: Remove the word “first”.

Ln 113: SI unit for gram is g

Ln 132: ”extract then”; the verb is missing: extract was then

Ln 145: 9700 rcf corresponds to what xg?

Ln 149: remove bracket; what was the standard for the calibration curve? Phytol? Please specify, as well as the concentration range. Describe better the samples for quantification of the total and of the unentrapped (supernatant) extract

Ln 166: remove colon

Ln 212: “plasma serum”. Which one? From previous sections, only plasma was stored.

ml and mL were used interchangeably. Please, homogenize. In general, homogenize units: µL and microliters; min and minutes, etc; Please, add a space between the value and the unit: eg. 37 ºC or 412 nm.

Ln 252: antioxidant: remove hyphen; At least 3 more occurrences

Fig.2: TEM picture: bar size is not readable

3.2- Only one batch was prepared? Is it reproducible? For the characterization and for the animal model a large amount of sample should have been prepared. Please provide mean and deviation values of structural and encapsulation parameters

Ln 277: Remove square brackets. Use normal brackets for introducing figures and table at the end of the sentence. In general, use normal brackets for tables and figures. Use square brackets for references only.

Figures 3-5: No statistical significance is presented. Use g instead of gm in the vertical axis legends

Table 3: Use g instead of gm

Ln 322: “Data are presentenced..” Please correct. Correct also in captions of figures 7, 8 and 9 due to copy/paste error

Figures 7-9: There is no need to write “Groups” bellow the group’s name in the xx axis.

Ln 474: Biochemical: use small caps

Please, check references format.

Reviewer 2 Report

The manuscript submitted by Mohammad Hossain Shariare, entitled « Liposomal Nano Drug Delivery of Blumea lacera Leaf Extract 2 and Its in-vivo Activity on CCl4 Induced Hepatotoxicity and 3 Oxidative Stress Rat aims to evaluate a liposomal nanoformulation containing Blumea lacera leaf extracts.

The manuscript has many weaknesses:

The authors state that « Blumea lacera (B. lacera) is a herbaceous plant belongs to the Asteraceae family, commonly found in south-east Asia. B. lacera is used traditionally as anti-diarrheal, antimicrobial, anxiolytic, anti-atherothrombosis, membrane stabilizing, anti-diabetic and alpha-amylase inhibitory activities. B. lacera also found to has antibiotic, antimicrobial and antioxidant effects [11-13]. Previous study showed that B. lacera leaf extract ameliorate acute ulcer and oxidative stress in ethanol induced rat model [14]”

If you use a complete extract, how can you be sure to target one way and not another?

The authors show a composition of the leaf extract. How to know which active ingredient in the mixture and in what quantity exactly will be encapsulated in the liposomes? What is the nature of these compounds? What is their stability at a very acidic pH (like that of the stomach)?

What is the release kinetics and of which compound?

Reviewer 3 Report

The article may be an useful contribution to the journal; however, few changes should be taken into consideration:

authors are advised to define all abbreviations at first appearance in text; in abstract and also elsewhere (e.g carbon tetrachloride, methods, etc)

In abstract must be explicitly stated what organisms the in vivo studies were performed on (e.g. Long Evan rats).

Choice of the number of rats should be explained, as 7 rats per individual group does not seem a large number.

Statistical issue: small number of rats/group does not justify use of parametric ANOVA; a non-parametric equivalanet of ANOVA should have been used, provided the data on such small number of rats is very less likely to be in all cases normal data; normlity tests results also missing; also, median should be reported, instead of average mean.

Also, repeated measures design checking for Time X Treatment interaction should be made with pairwise comparisons;

Table 2 and 3 should add more information such as p-values, or should have added similar charts as in Figure 6 for stressing the significance of difference between groups.

Grammar and punctuation must also be carefully checked within the entire article.

Round 2

Reviewer 2 Report

no comments.